# Ultrahigh-Q guided mode resonances in an All-dielectric metasurface

Lujun Huang [1,9] ✉, Rong Jin[2,3,4,9], Chaobiao Zhou[5,9], Guanhai Li[2,3,4] ✉, Lei Xu [6], Adam Overvig [7], Fu Deng[1], Xiaoshuang Chen[2,3,4], Wei Lu[2,3,4], Andrea Alù [7,8] ✉ & Andrey E. Miroshnichenko [1] ✉

High quality(Q) factor optical resonators are indispensable for many photonic devices. While very large Q-factors can be obtained theoretically in guided-mode settings, free-space implementations suffer from various limitations on the narrowest linewidth in real experiments. Here, we propose a simple strategy to enable ultrahigh-Q guided-mode resonances by introducing a patterned perturbation layer on top of a multilayer-waveguide system. We demonstrate that the associated Q-factors are inversely proportional to the perturbation squared while the resonant wavelength can be tuned through material or structural parameters. We experimentally demonstrate such high-Q resonances at telecom wavelengths by patterning a low-index layer on top of a 220 nm silicon on insulator substrate. The measurements show Q-factors up to $2.39 \times 10^5$, comparable to the largest Q-factor obtained by topological engineering, while the resonant wavelength is tuned by varying the lattice constant of the top perturbation layer. Our results hold great promise for exciting applications like sensors and filters.

High-Q photonic nano-resonators constitute a core element in high-performance optoelectronic and photonic devices in modern optical communications. While very large Q-factors have been obtained with micro-ring resonators and micro-disks via the optical fiber excitation through in-plane near-field coupling[1], it is very challenging to achieve an ultrahigh Q-factor in experiments via free-space excitation. Resonant waveguide gratings and photonic crystal slab are viewed as an ideal platform to achieve high-Q guided-mode resonances (GMRs)[2–4], which can be accessed in free space. They have been widely used in in filtering[5–10], sensing[11–13], and wavefront shaping[14,15]. Although the Q-factors of GMRs in theory can be very high, the largest Q-factor in experiments has been so far limited by fabrication imperfections (i.e.,

roughness, disorder and non-uniformity) causing increased radiation loss. The nanofabrication process usually involves dry-etching to define these nanostructures, introducing additional surface roughness[16], imperfect vertical side walls[17], and long-range non-uniformity[18]. To date, most of the reported Q-factors of GMRs in experiments fall in the range of from few hundreds to thousands, as summarized in Table 1. High-Q resonances up to 2340 have been realized with plasmonic metasurfaces[19] by harnessing surface lattice resonances, but such plasmonic approaches are widely recognized to be limited by intrinsic material loss. Another candidate to achieve high-Q resonances is the all-dielectric metasurface exploiting electromagnetically induced transparency phenomena[20].

[1]School of Engineering and Information Technology, University of New South Wales, CanberraNorthcott DriveACT 2600, Australia. [2]State Key Laboratory of Infrared Physics, Shanghai Institute of Technical Physics, Chinese Academy of Sciences, 500 Yu Tian Road, Shanghai 200083, China. [3]Hangzhou Institute for Advanced Study, University of Chinese Academy of Sciences, No.1 SubLane Xiangshan, Hangzhou 310024, China. [4]Shanghai Research Center for Quantum Sciences, 99 Xiupu Road, Shanghai 201315, China. [5]School of Physics and Mechatronic Engineering, Guizhou Minzu University, Guiyang 550025, China. [6]Advanced Optics and Photonics Laboratory, Department of Engineering, School of Science Technology, Nottingham Trent University, Nottingham NG11 8NS, UK. [7]Photonics Initiative, Advanced Science Research Center, City University of New York, New York, NY 10031, USA. [8]Physics Program, Graduate Center, City University of New York, New York, NY 10016, USA. [9]These authors contributed equally: Lujun Huang, Rong Jin, Chaobiao Zhou.
✉ e-mail: ljhuang@mail.sitp.ac.cn; ghli0120@mail.sitp.ac.cn; aalu@gc.cuny.edu; andrey.miroshnichenko@unsw.edu.au

**Table 1 | Summary of measured Q-factor in the grating, photonic crystal slab, and metasurface**

| Mechanism | Q | λ(nm) | $\theta_{inc}$(°) | Structure | Substrate | Etching (Y/N) | Sample size (µm²) | Ref |
|---|---|---|---|---|---|---|---|---|
| SLR | 2340 | 1550 | 0 | Au NPs Array | Silica glass | N | 600 × 600 | 19 |
| EIT | 483 | 1371 | 0 | Si metasurface | Quartz | Y | 225 × 225 | 20 |
| GMR (TE) | 8000 | 750 | 0 | SiO₂ grating On Si₃N₄ | Glass | Y | 10,000 × 15,000 | 12 |
| GMR (TM) | 4500 | 805 | 0 | | | | | |
| GMR | 2700 | 1304 | 0 | Shallow Si grating | SiO₂ | Y | 5000 × 5000 | 6 |
| GMR | 391 | 860 | 0 | Resist grating on HfO₂ | Fused-Silica | N | – | 5 |
| GMR | 306 | 1531 | 45 | SiO₂ grating | SOI | Y | 5000 × 7000 | 13 |
| GMR | 32,000 | 490 | 0.2 | Si₃N₄ PCS | 6 µm SiO₂ on Silicon | Y | >10,000 × 10,000 | 4 |
| GMR | 2500 | 1465 | 0 | Si metagrating | Sapphire | Y | 300 × 300 | 14 |
| SP-BIC | 1946 | 1425 | 0 | Si metasurface | SOI | Y | 410 × 410 | 44 |
| SP-BIC | 10,000 | 583 | 0.1 | Si₃N₄ PCS | SiO₂ | Y | 17,300 × 17,300 | 46 |
| SP-BIC | 2750 | 825 | 4 | GaAs metasurface | SiO₂ | Y | 60 × 108 | 29 |
| BIC | 18,511 | 1588.4 | 0 | Si metasurface | SiO₂ | Y | 19 × 19 | 37 |
| Resonance trapped-BIC | 4700 | 1551.4 | 0 | InGaAsP metasurface | No Substrate | Y | 19 × 19 | 26 |
| Topological BIC | 490,000 | 1568.3 | 1.2 | Si PCS | No Substrate | Y | 250 × 250 | 50 |
| Topological BIC | 7250 | 1595 | - | InGaAsP PCS | No Substrate | Y | 16 × 16 | 30 |
| **GMR** | **239,000** | **1551** | **0** | **Resist PCS** | **220 nm SOI** | **N** | **520 × 520** | **This work** |

Recently, bound states in the continuum (BICs) have introduced a viable alternative to realize ultrahigh-Q resonances in a free space setting[21–25]. The giant field enhancement typically associated with large-Q factors can benefit applications ranging from lasing[26–30], sensing[31,32], strong coupling[33,34] to enhanced nonlinear harmonic generations[35–38]. Most of the quasi-BICs(QBICs)-based high-Q modes reported so far are achieved by patterning the high-index semiconductor thin film as a single nanoparticle[39–41] or an array of nanoparticles on substrate[21,22,26,29,37,42–46]. Table 1 summarizes the measured Q-factor of QBIC modes based on all-dielectric metasurfaces. Typically, the Q-factors of QBICs are limited by scattering loss caused by unavoidable fabrication imperfections (i.e., roughness, disorder). Also, the presence of a substrate may also introduce the additional leaky channel due to out-of-plane asymmetry, further reducing the Q-factors of QBICs. Of course, ultrahigh-Q modes can still be constructed by carefully arranging the structure parameters. For example, topological BICs, arising from merging multiple BICs into a single one, are found to be less sensitive to fabrication imperfections[47–52]. However, achieving such a high Q-factor requires delicate engineering of the structure parameters and may not always be compatible with other design goals. Moreover, the substrate must be removed to satisfy the requirement for environmental symmetry. In addition, generalizing this achievement to visible wavelengths is an outstanding challenge that may prove difficult due to materials and fabrication constraints. To promote widespread adoption in practical applications, it is highly desirable to develop a universal yet simple design strategy to realize ultrahigh Q-resonances that minimize the impact of fabrication imperfections, mitigate the substrate effect, and do not need careful engineering based on complex topological concepts.

In this work, we introduce a strategy to implement ultrahigh-Q guided-mode resonances in an all-dielectric metasurface without the need for topological engineering of the structure or introducing any symmetry breaking other than the discretization of translational symmetry. Instead of patterning the high-index layer to form a Mie resonator in a unit cell, we introduced an ultrathin photoresist layer as a perturbation layer on top of a multilayer-waveguide system so that guided modes are transformed into leaky modes that produce GMRs, which obviates the need for dry-etching. High Q-factors of GMRs can be easily realized as they are inversely proportional to the perturbation parameter squared ($Q \propto \alpha^{-2}$). The perturbation can be reduced to a very small value because the fabrication of such a device only involves

spin-coating of resist and developing, enabling minimal perturbation by removing the roughness and disorder of the sample. The validity of such a design methodology is confirmed by patterning a low-index photoresist thin film on a standard silicon on insulator (SOI) sample that serves as a simple waveguide system. The experimental results show that the Q-factor was as high as $2.39 \times 10^5$, comparable to the maximum Q-factor of topological BICs. In addition, the resonant wavelength was easily tuned by changing the period of the meta-waveguide system. This design strategy is robust and powerful in obtaining ultrahigh-Q resonances at arbitrary operating wavelengths because it can be generalized to any other waveguide system made of any different high refractive index ($n = 2\sim5$) as the core layer. Our results may find great potential in applications requiring very sharp spectral features, such as sensing and filtering.

## Results

### Guided modes in a multilayer-waveguide system

We start by investigating the leaky modes of a multilayer waveguide structure, consisting of a high refractive index $n_2$ layer with finite thickness sandwiched between two semi-infinite layers with low refractive index being $n_1$ and $n_3$, respectively. Without loss of generality, we assume that the electric field component is perpendicular to the plane of structure, which is E//y. Such an open system supports a series of Fabry-Perot resonant modes with low-Q factors (See Section 1 of supplementary materials and Fig. S1). If we apply a virtual periodic boundary condition with arbitrary periods, in addition to the low-Q leaky modes, such a simple geometry can also support guided modes (GMs) with an infinite Q-factor. Note that there are two ways of defining a periodic structure: a 1D grating or a 2D metasurface, as shown in Fig. 1a and Fig. S2. The band diagram for TE (E//y) and TM (H//y) modes can be obtained by folding the waveguide dispersion of GMs into the first Brillouin zone. The dispersion relationship of GMs has been derived in the text book[53] and can be expressed as follows

$$\text{TE}: \tan(\beta_2 t) = \frac{\beta_2(\beta_1 + \beta_3)}{\beta_2^2 - \beta_1\beta_3} \quad (1)$$

$$\text{TM}: \tan(\beta_2 t) = \frac{n_2^2\beta_2(\beta_1 + n_3^2\beta_3)}{n_3^2\beta_2^2 - n_2^2\beta_1\beta_3} \quad (2)$$

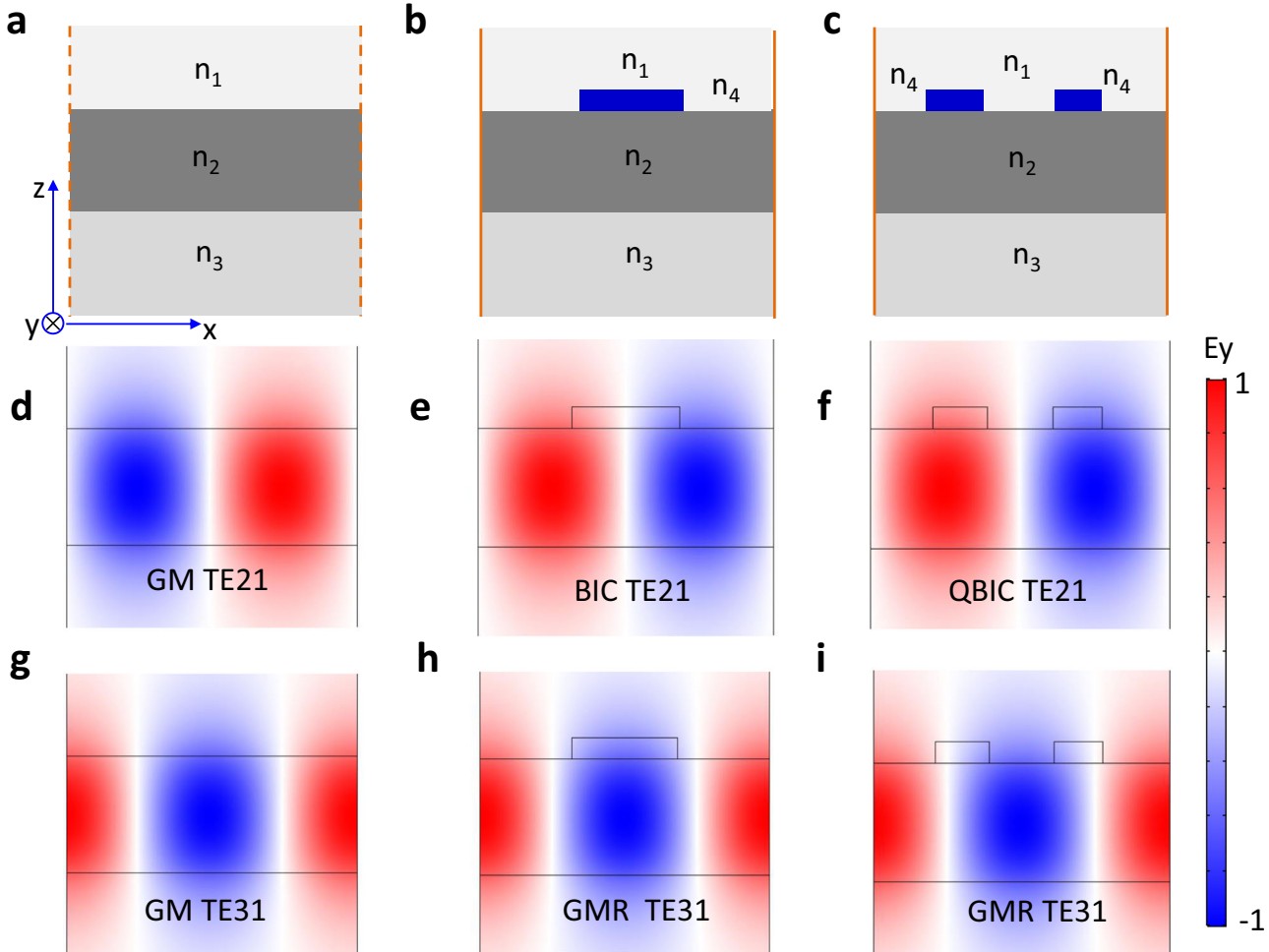

**Fig. 1 | Designing strategy of high-Q GMRs. a–c** Schematic drawing of a three-layer waveguide system (**a**), and meta-waveguide system with a top perturbation layer as simple grating (**b**) and compound grating (**c**) on the top. **d**, **g** are Ey distribution of GMs TE$_{21}$ and TE$_{31}$. **e**, **h** are Ey distribution of symmetry protected BIC TE$_{21}$ and GMR TE$_{31}$ in the standard grating. **f**, **i** are Ey distribution of QBIC TE$_{21}$ and GMR TE$_{31}$ for the compound grating.

Where $\beta_1 = \sqrt{k^2 - n_1^2 k_0^2}$, $\beta_2 = \sqrt{n_2^2 k_0^2 - k^2}$, $\beta_3 = \sqrt{k^2 - n_3^2 k_0^2}$, $k$ is the propagating constants of guided mode, $k_0 = \omega/c$, $\omega$ is the angular frequency and c is the speed of light in the vacuum.

It is necessary to mention that there is a cut-off frequency for different guided modes.

$$\Omega_{cj} = \frac{c}{t\sqrt{n_2^2 - n_3^2}}(\tan^{-1}\left[s\sqrt{\frac{n_3^2 - n_1^2}{n_2^2 - n_3^2}}\right] + j\pi)(j = 0,1,2\ldots) \quad (3)$$

Where $s = 1$ for TE and $s = n_2^2$, c is the speed of light in the vacuum.

More detailed discussions on band structure calculation are given in section 2 of supplementary materials and Fig. S3.

## Guided mode resonances and BICs

Here, we mainly focus on the GMs at Γ-point in the first Brillouin zone in momentum space. Alternatively, the eigenfrequencies of these GMs can be calculated quickly with the commercial software COMSOL Multiphysics. For the 1D grating system, these GMs are designated as TE$_{ml}$ (TM$_{ml}$)[54,55], where m and l are the numbers of antinodes of the electric (magnetic) field in the x and z directions, respectively. Fig. 1b, c shows two typical examples of GMs: modes TE$_{21}$ and TE$_{31}$. These GMs can be transformed into GMRs by introducing an ultrathin low-index metasurface on top of the high-index layer. For example, when the top perturbation layer is patterned as a grating, GMs TE$_{21}$ and TE$_{31}$ are

converted into symmetry-protected BIC mode TE$_{21}$ and GMR TE$_{31}$ with a finite Q-factor, respectively. If the mirror symmetry of the top periodic structure is broken with respect to the z-axis, BIC mode TE$_{21}$ is reduced to QBIC. Note that GMs like TE$_{21}$ and TE$_{31}$ at the Γ point share a similar feature with symmetry protected BICs. When the periodic perturbation is introduced in the multilayer waveguide system, such GMs with infinite Q-factors are successfully converted into either a BIC mode with an infinite Q-factor or a GMR with a finite Q-factor depending on the mode parity. If the perturbation approaches zero, the radiative decay rates of GMRs are reduced to zero. Rigorous derivation shows that Q values are inversely proportional to β² (see section 3 of supplementary materials), where β is the perturbation parameter of the system. This is similar to the symmetry-protected BICs with Q ∝ α⁻² (where α is the asymmetry parameter of unit cell). The GMs for the TM case are shown in Fig. S4. The following sections demonstrate that GMs provide much freedom and flexibility to design ultrahigh Q-factor resonance with a subwavelength meta-waveguide system at an arbitrary operating wavelength.

For simplicity, we use a three-layer structure air/Si/SiO$_2$ to design a high-Q resonance at $\lambda_0 = 1550$ nm. Without loss of generality, the thickness of the middle layer Si is chosen as 220 nm. The refractive index of Si and SiO$_2$ are set as 3.47 and 1.46, respectively. The dispersion relationship of GMs in such a three-layer system can be obtained from Eq. (1) and is shown in Fig. 2. To support a GM at $\lambda_0 = 1550$ nm (dashed black line in Fig. 2), the propagation constants should

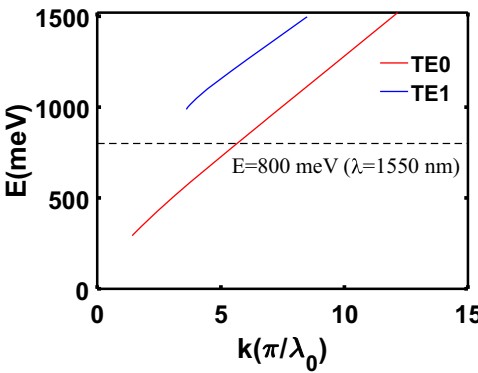

**Fig. 2 | Dispersion relationship of GMs in a multilayer system (air/220 nm Si/ SiO₂).** Solid blue and red lines corresponds to GMs TE0 and TE1, respectively. The dashed black line indicates the GM wavelength at 1550 nm.

be $k = 5.6515 * \pi/\lambda_0$. Then, the virtual period can be obtained as $p = 548$ nm by applying $k = 2\pi/p$. In principle, the resonant wavelength can be tuned to any other value by varying the period, which is discussed in the later section.

By introducing a top perturbation layer that is patterned as either a grating or a metasurface, the GM evolves into a BIC or GMR depending on the structure's geometry and the mode's parity. Their band structures are plotted in Figs. S5, 6. Here, it is worth noting that the refractive index of grating or metasurface can be of any value as long as the top layer serves as a perturbation layer. As an example, the refractive index of the top layer is set as 1.46, matching the index of SiO₂. We first consider a simple grating structure shown in Fig. 3a and study the effect of width and thickness on the Q-factor of GMR. From Fig. 3b, it can be found that the Q-factor of the GMR TE$_{31}$ is proportional to $t^{-2}$ when the width of the grating is fixed as 200 nm. Excellent agreement can be found between the calculated Q-factor (solid red line) and fitted Q-factor (dashed blue line) by applying the fitting equation $Q = Ct^{-2}$ (C is the fitted constant). This is similar to the observation in symmetry-protected BICs, where $Q \propto \alpha^{-2}$ (where $\alpha$ is the asymmetry parameter). Such a relationship is well explained by perturbation theory[56] (see section 3 in supplementary materials). This is also confirmed by the reflection spectrum mapping as the thickness increases from 0 to 300 nm (See Fig. S7).

Similarly, the Q-factor shows a linear dependence on $w^{-2}$ or $(d-w)^{-2}$ for a given thickness, as confirmed in Fig. 3c. Thus, it can be safely concluded that the effective perturbation area or volume plays the dominant role in governing the Q-factors of GMRs. Moreover, the resonant wavelength remains almost the same with the changing thickness or width because of the low-index nature of the top perturbation layer (See Figs. S7, 8). As described in Fig. 1, except for the GMR TE$_{31}$, such a grating also supports a symmetry-protected BIC mode TE$_{21}$ due to structural symmetry. Further breaking the symmetry can induce the transition from BIC to QBIC. For instance, by introducing an asymmetric air slit inside the rectangular nanowire, BIC mode TE$_{21}$ is successfully converted into the QBIC. Fig. 3b shows the Q-factor of QBIC TE$_{21}$ as a function of thickness. The case of TE$_{31}$ is also plotted as a reference to make a comparison in Fig. 3d–f. Similar to the case of TE$_{31}$ shown in Fig. 3b, the Q-factor of QBIC TE$_{21}$ is proportional to $t^{-2}$ when an asymmetric slit is introduced with $g = 120$ nm and $xc = 10$ nm. The left and right nanowires' widths are 110 nm and 90 nm, respectively. Besides, we find that the Q-factor of QBIC mode TE$_{21}$ is more than one order of magnitude higher than that of mode TE$_{31}$. This is understandable because the perturbation required for symmetry-breaking is much smaller for QBIC TE$_{21}$ compared to GMR TE$_{31}$. Such a general conclusion also can be applied to the 3D metasurface. When the cuboid unit cell of the top perturbation layer is arranged as a square lattice, such a metasurface supports degenerate symmetry-protected

BICs (TE$_{211}$ and TE$_{121}$) and GMRs (TE$_{311}$ and TE$_{131}$). The degeneracy could be easily lifted by choosing the different lattice constants along the x and y-axis (See Fig. S9). Further introducing in-plane broken symmetry leads to the transition from BICs into QBICs for the TE$_{211}$ mode. Fig. 3g–i shows the Q-factor of GMR TE$_{311}$ and QBIC TE$_{211}$ versus the thickness of top layer. Additionally, it is not surprising that Q-factor follows a similar trend as 1D grating, $Q \propto t^{-2}$. Moreover, such a 2D metasurface supports BICs or GMRs for TM cases (See Fig. S10). Here, it is necessary to point out that the top layer can also be patterned as a rectangular lattice with any other shape such as photonic crystal slab because it just functions as a weak perturbation layer.

## Guided mode resonances engineering

Note that many GMs can be easily constructed with such a simple meta-waveguide system. We again consider the 1D grating structure shown in Fig. 1a with material parameters and geometry parameters given in Fig. 3a. Except for TE$_{21}$ and TE$_{31}$ mode, such a three-layer structure supports other high-order GMs. Fig. 4a shows the field distributions of six GMs: TE$_{21}$ and TE$_{31}$, TE$_{22}$ and TE$_{32}$, TE$_{41}$ and TE$_{51}$. Their resonant wavelengths are presented in Fig. 4b. Note that all of these modes are dominated by the zero-order diffraction. Of course, even higher-order GMs like TE$_{42}$ and TE$_{52}$ can be found. However, their Q-factor may decrease significantly due to the enhanced radiation from unwanted diffraction after the perturbation is introduced. Also, GMs may be designed at any pre-defined wavelength with different structure parameters and material parameters. For example, if the thickness of the Si layer is set as 220 nm or 340 nm, we can always tune the virtual period so that the resonant wavelengths of GM TE$_{21}$ or TE$_{31}$ could cover a broadband wavelength range from 1000 nm to 1800 nm, as displayed in Fig. 4c.

Similarly, from Fig. 4d, we found that the resonant wavelength can be tuned by varying the thickness of Si layer while the period is fixed as 450 nm (or 550 nm). One can also engineer the GM's wavelength by choosing materials with different refractive indices. For example, to design a GM at 1550 nm, the period should be varied from 1000 nm to 460 nm when the refractive index $n_2$ increases from 2 to 4. The corresponding result is shown in Fig. 4e. To verify the robustness of such a design strategy in realizing ultrahigh-Q resonances, we calculate the Q-factor of GMR TE$_{31}$ in a grating system for the middle layer with various refractive index $n_2$. Indeed, all of the Q-factors decrease significantly with the increasing thickness of the top grating layer, as illustrated in Fig. 4f. Nevertheless, it is interesting that the Q-factor for high-index $n_2 = 3.5$ is almost one order higher than the low-index $n_2 = 2.0$. This can be intuitively understood by the reduced index contrast between the substrate (or superstrate) and the sandwiched high-index layer, which leads to more energy radiation into the substrate (or superstrate) instead of confining inside the high-index layer. The robustness of such a design strategy could enable us to design ultrahigh-Q GMRs at ultraviolet and visible wavelength ranges by choosing appropriate core layers with high-index, including Si₃N₄, GaN, TiO₂. In the supplementary materials, we demonstrate two examples of designing high-Q resonators in the visible wavelength by using GaN and Si₃N₄ as core layer, as shown in Figs. S11, 12.

## Experimental demonstrations of high-Q GMRs

After gaining a solid understanding of designing ultrahigh-Q resonances from GMs, we move to the experimental demonstration of such a high-Q resonance. Fig. 5 shows the experimental setup of a custom-made crossed-polarization measurement system. Crossed polarization measurement has been widely used to measure the high-Q resonances[50,57]. We fabricate a series of photonic crystal slabs made of 330 nm thick photoresist (ZEP520) on an SOI substrate (220 nm-Si/2 μm-SiO₂/Si) using electron-beam lithography followed by developing process. Here, the photo-resist layer instead of the SiO₂ layer is patterned as a perturbation layer. On the one hand, the photoresist has a

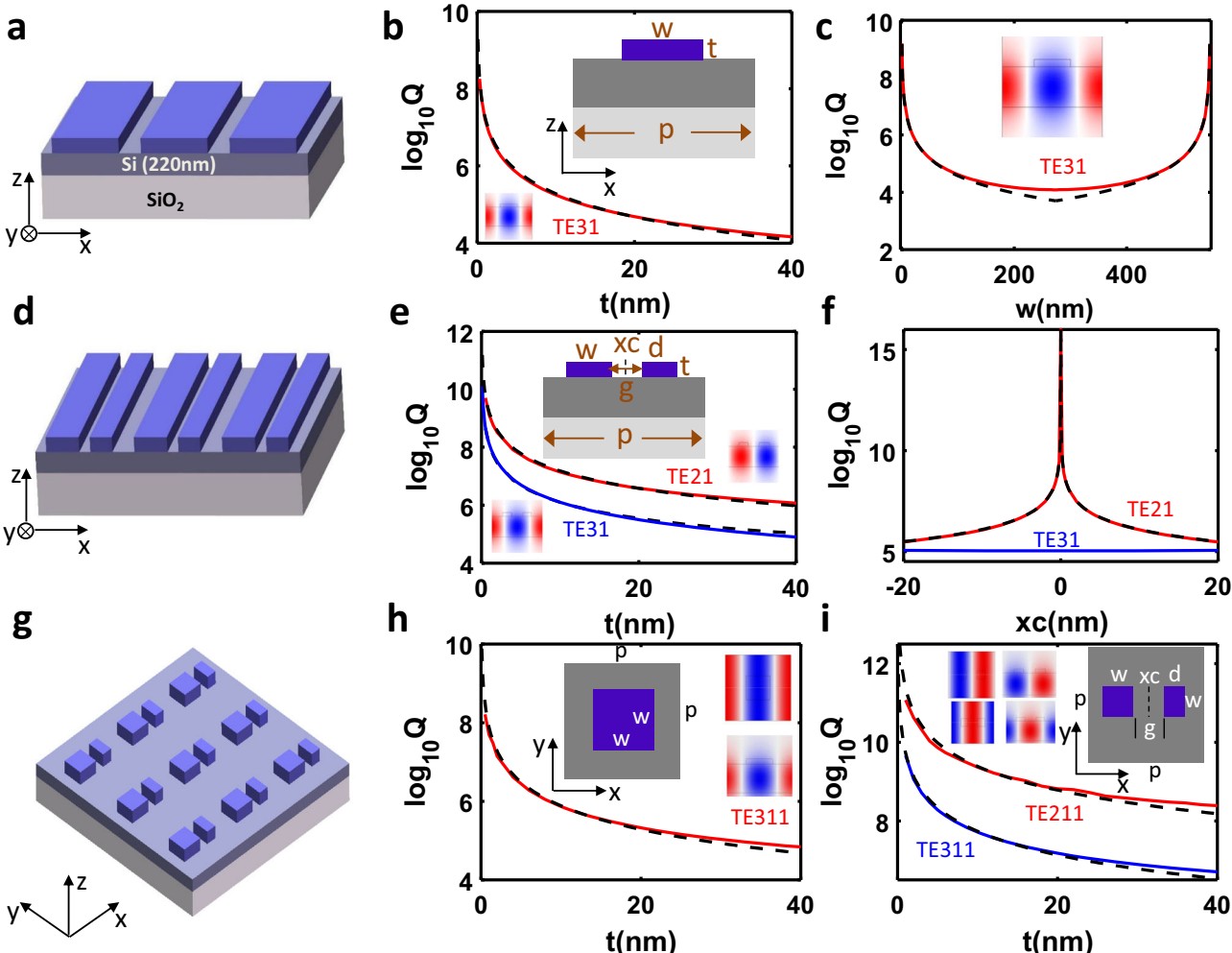

**Fig. 3 | High-Q GMRs. a** Schematic drawing of a meta-waveguide system made of an ultrathin SiO₂ grating on top of 220 nm-Si on SiO₂ substrate. **b** The Q-factor of GMR mode TE$_{31}$ vs thickness of top perturbation layer. Solid and dashed lines refer to the numerically calculated Q-factor and fitting Q-factor. The structure parameters are $p = 548$ nm and $w = 200$ nm. **c** The Q-factor of GMR mode TE$_{31}$ vs grating's width. Solid and dashed lines show the numerically calculated Q-factor and fitting Q-factor. The structure parameters are $p = 548$ nm and $t = 40$ nm. **d** Schematic drawing of a meta-waveguide system made of an ultrathin SiO₂ compound grating on top of 220 nm-Si on SiO₂ substrate. **e** The Q-factor of QBIC TE$_{21}$ (solid red) and GMR TE$_{31}$ (solid blue) vs thickness of top perturbation layer. Dashed lines are fitting Q-factor. The structure parameters are $p = 548$ nm, $w = 110$ nm, $d = 90$ nm,

$g = 120$ nm, xc = 10 nm. **f** The Q-factor of BIC TE$_{21}$ (solid red) and GMR TE$_{31}$ (solid blue) vs gap center xc. The structure parameters are $p = 548$ nm, $w + d = 200$ nm, $g = 120$ nm. Dashed lines are fitting Q-factors. **g** Schematic drawing of a meta-waveguide system made of an ultrathin SiO₂ metasurface on top of 220 nm-Si on SiO₂ substrate. **h** The Q-factor of GMR mode TE$_{311}$ vs the thickness of top perturbation layer. Solid and dashed lines show the numerically calculated Q-factor and fitting Q-factor. The structure parameters are $p = 548$ nm, $w = 200$ nm. **i** The Q-factor of BIC TE$_{211}$ (solid red) and GMR TE$_{311}$ (solid blue) vs top layer thickness. Dashed lines are fitting Q-factor. The structure parameters are $p = 548$ nm, $w + d = 200$ nm, $g = 120$ nm.

low refractive index of around 1.4. On the other hand, no other post-processing (i.e., dry etching) is applied to the other layers so that higher sample quality is guaranteed.

Fig. 6a shows the schematic drawing of the whole device, and Fig. 6b shows the scanning electron microscopy image of one typical fabricated sample. The lattice constants along the x- and y-axis are identical in this device. We first investigate the effect of hole size on the Q-factor by fixing the lattice constant as 545 nm but varying the hole radius. Eigenmode calculations indicate that such a structure can support degenerate GMRs TE$_{311}$ and TE$_{131}$ at around 1550 nm. It also supports symmetry protected BICs TE$_{211}$ and TE$_{121}$. However, they cannot be excited at normal incidence unless the in-plane mirror symmetry is broken with respect to the y-z plane.

The relevant results are put in Fig. 6d–f. It can be observed from Fig. 6d that shrinking the size of the hole on the top perturbation layer leads to the narrowing of resonance linewidth and slight redshift of resonance. We retrieved the Q-factors and resonant wavelengths of

these high-Q resonances by fitting them to a Fano-profile. Fig. 4e shows one example of Fano fitting for $R = 72$ nm. The measured Q-factor for structures with different hole sizes is presented in Fig. 6f, where the highest Q-factor is up to $2.39 \times 10^5$ at $R = 72$ nm. Such a high Q-factor is comparable to the record-high Q-factors in a topological metasurface[50]. Nevertheless, our approach involves only patterning the top resist layer without the need for carefully engineering the topological features. Also, the calculated Q-factor is plotted as a reference and is found to show an excellent agreement with the experiment measurement. In fact, we also fabricate the sample with an even smaller radius $R = 35$ nm. However, it becomes very challenging to get an even higher Q-factor in experiment because the peak cannot be distinguished from the noise or background. Further optimization of the measurement system is needed. Such an ultrahigh-Q factor can be attributed to two factors: small hole radius and shallow hole with partial unetched resist film underneath. After taking both factors into account, the calculated Q is an order of million for such a size (See Fig. S13).

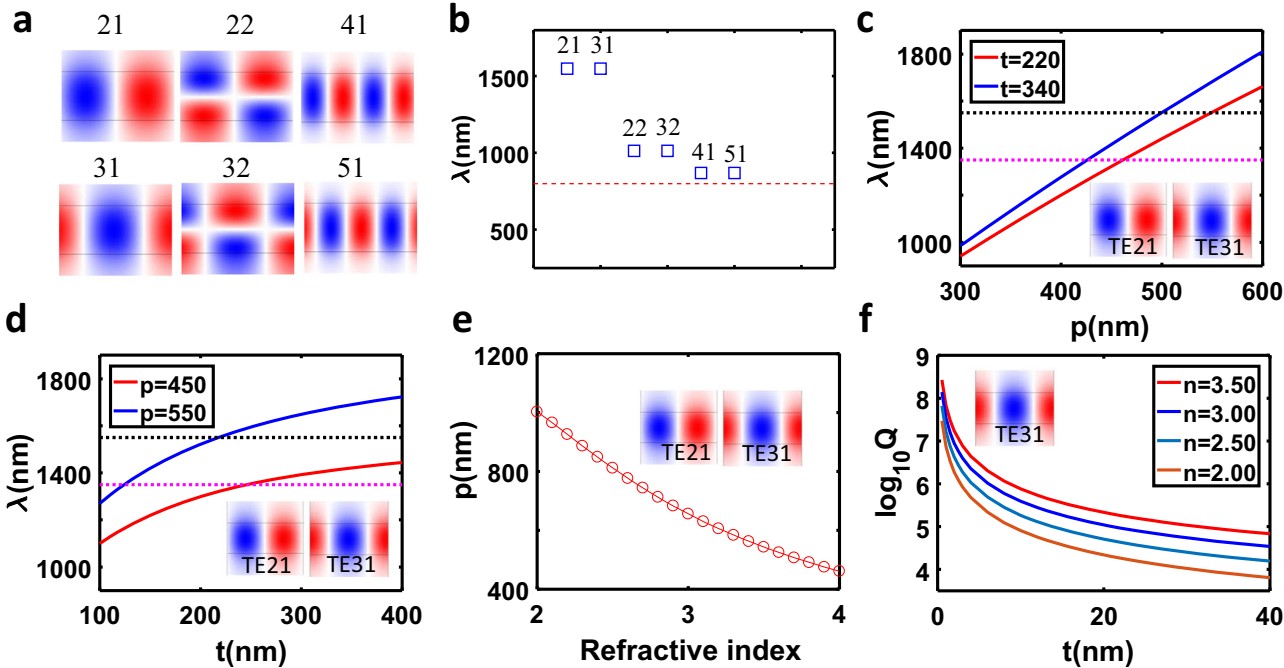

**Fig. 4 | Designing GMs at arbitrary wavelengths. a** Eigenfield distributions Ey of six GMs in the 220 nm-Si on SiO$_2$ substrate with virtual period $p$ = 548 nm. **b** Resonant wavelengths of six GMs. The dashed line indicates the first-order diffraction wavelength. **c** Resonant wavelength engineering through the virtual period for GMs TE$_{21}$ and TE$_{31}$. The dashed lines indicate the resonant wavelength 1550 nm (black) and 1350 nm (magenta). **d** Resonant wavelength engineering through the thickness of Si layer for GMs TE$_{21}$ and TE$_{31}$. The dashed lines indicate the resonant wavelength 1550 nm (black) and 1350 nm (magenta). **e** Virtual period of three-layer waveguide as a function of refractive index of middle layer for GMs TE$_{21}$ and TE$_{31}$. Here, the resonant wavelengths of two GMs are fixed as 1550 nm. **f** Q-factors of GMR TE$_{31}$ vs the thickness of top perturbation layer for the meta-waveguide system with varied refractive index of middle layer.

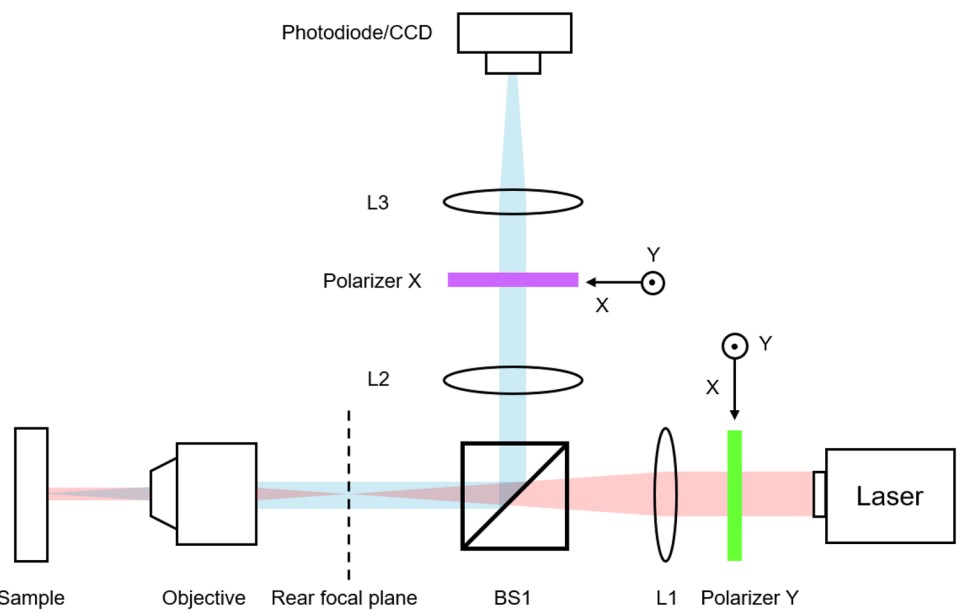

**Fig. 5 | Schematic illustration of experimental setup.** Light red and light blue lines indicate the incident light and direct reflection from the sample, respectively. L means lens and BS means beam splitter.

Finally, to demonstrate the tunability of resonant wavelength, we fabricate five samples with the period varying from 535 nm to 555 nm while their hole radii are fixed as 80 nm. The scattering spectra are plotted in Fig. 6g. Indeed, the resonant peaks show a linear dependence on the period (Fig. 6h), matching very well to the theoretical calculation. In addition, all of the measured Q-factor are ranged between $1.0 \times 10^5$ and $2.0 \times 10^5$ (Fig. 6i). Note that TE$_{311}$ and TE$_{131}$ are degenerate for the identical lattice constants along the x and y-axis. This degeneracy can be lifted by employing a rectangular lattice with different $p_x$ and $p_y$. Thus, high-Q resonance could be implemented at a different wavelength by switching the polarization (See Fig. S14). In other words, even a single structure can support two high-Q modes at different wavelengths for x and y-polarization. An even higher-Q factor can be obtained by exciting QBICs via breaking the unit-cell's mirror

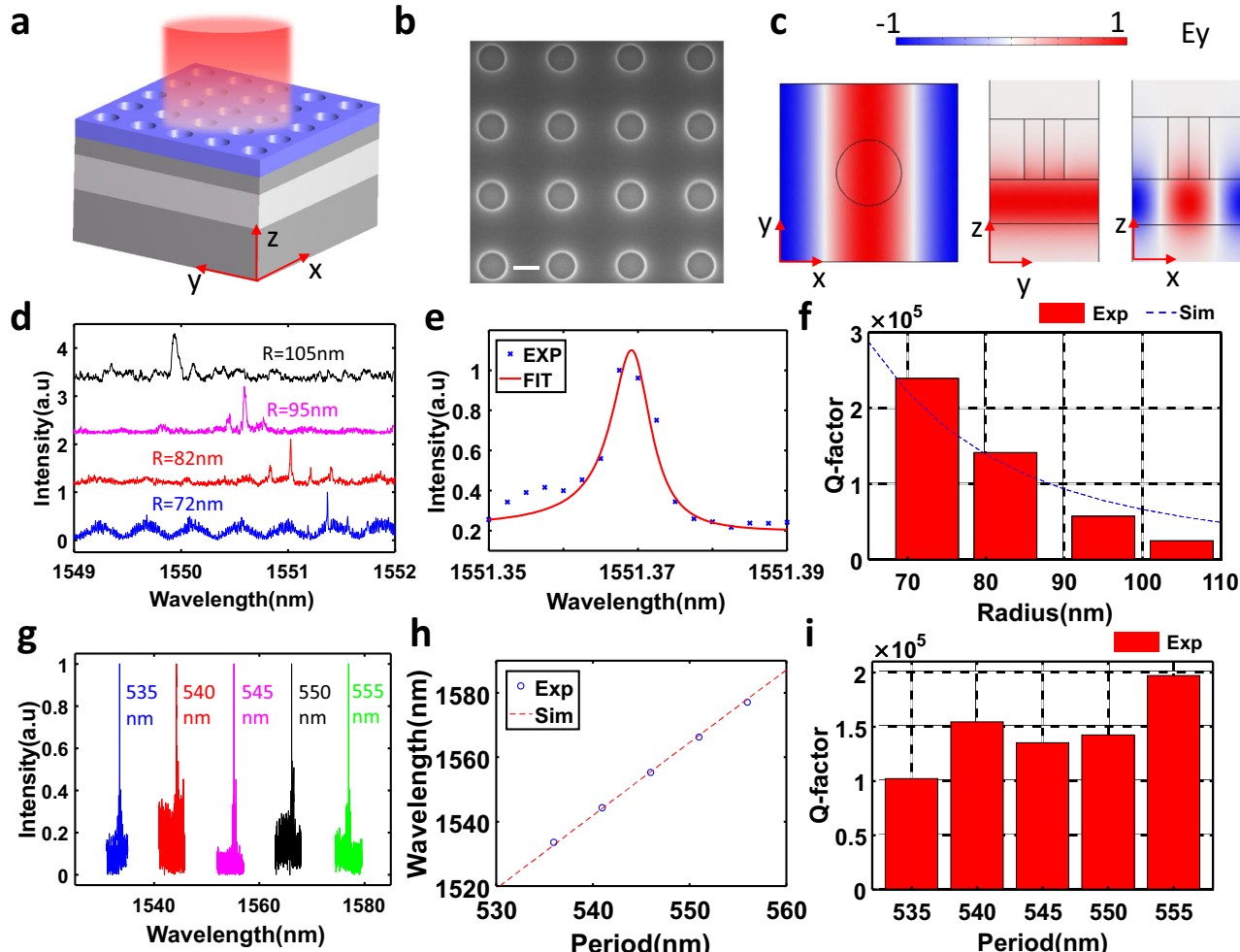

**Fig. 6 | Experimental demonstration of high-Q resonances based on GMs.**
**a** Schematic drawing of a 3D meta-waveguide system. The top perturbation is patterned as photonic crystal slab with air hole. **b** SEM image of a fabricated meta-waveguide system. The scale bar in white is 200 nm. **c** Eigenfield distribution for GMR $TE_{3||}$. **d** Measured reflection spectra for four fabricated samples with fixed period 545 nm but varied hole radius from 72 nm to 105 nm. **e** Fano-fitting for reflection spectrum for hole radius $R = 72$ nm. **f** Retrieved Q-factors vs radius for mode $TE_{3||}$, dash line is the simulated Q-factor. **g** Measured reflection spectra for a meta-waveguide system when periods change from 535 nm to 555 nm. **h** Comparison between calculated and measured resonant wavelength. **i** Retrieved Q-factors vs period. a.u in the figure the abbreviation of arbitrary units.

symmetry, as demonstrated in Fig. 3i. In addition to reducing the hole size, such an ultrahigh-Q mode may be enabled by using an ultrathin material as a perturbation layer. The recently developed two-dimensional (2D) materials, especially 2D transition metal dichalcogenides (TMDCs)[58,59] and hexagonal boron nitride (hBN)[60], are ideal candidates for realizing atomically thin film below 10 nm. Moreover, such a 2D TMDC multilayer with thickness below 10 nm can be easily patterned by induced coupled plasma (ICP) etching. Although the refractive index of TMDC is a bit high ($n = 3\sim4.5$ in the near-infrared), their atomically thin nature makes them an excellent candidate as a perturbation layer. We expect a high-quality thin film like TMDCs and hBN multilayers, and mature fabrication process could help to realize an ultrahigh-Q factor larger than $10^6$ or even more.

Note: The results presented in this work are in bold. EIT-electromagnetic induced transparency; SLR- surface lattice resonance; GMR-guided mode resonance; BIC-bound state in the continuum; NP-nanoparticle; PCS-photonic crystal slab; SOI-silicon on insulator.

## Discussion

In summary, we theoretically proposed and experimentally demonstrated high-Q guided resonances in a meta-waveguide system. The designing strategy builds upon adding a perturbation layer (i.e., photoresist) on top of a conventional waveguide system, which converts GMs into the BICs or GMRs. The Q-factors of GMRs strongly depend on the perturbation of a meta-waveguide system, suggesting a broadly adaptable way of realizing ultrahigh-Q resonances without resorting to topological concept, as demonstrated in the recent literature [46]. The ultrahigh Q-factor of GMRs can be realized because the fabrication of proposed structure only involves resist-coating and developing, but no deep etching process, successfully eliminating the common source of roughness and disorder of the sample and enabling a controlled tiny perturbation in realistic devices. We experimentally demonstrated the feasibility of this design strategy in achieving high-Q resonances at around 1.55 μm by fabricating a thin photoresist layer photonic crystal slab on top of 220 nm Si/2 μm SiO$_2$/Si (standard SOI sample). The measured Q-factor reaches up to $2.39 \times 10^5$, comparable to the value of merging BICs by topological charge engineering. Also, the resonant wavelength can be tuned by varying the lattice constant of the top resist layer. Furthermore, by choosing different lattice constants along the x- and y-axis, we realized polarization-dependent GMRs. Our findings open a new avenue to design optoelectronic and photonic devices in which ultra-sharp spectral features may improve their performance, such as biosensors and filters.

## Methods

### Numerical simulation

The eigenmodes of meta-waveguide are calculated by commercial software Comsol Multiphysics based on finite-element-method (FEM). The reflection and transmission spectra of meta-structures are calculated by rigorous-coupled wave analysis (RCWA).

### Fabrication

Our devices were fabricated on a silicon-on-insulator (SOI) wafer with a 220 nm silicon layer and 2 µm-thick buried layer. The SOI was firstly coated with the 330 nm-thick resist layer (ZEP520). Then, the device patterns were defined on the resist by electron-beam lithography. After that, the devices were carefully developed and fixed by dimethyl benzene and iso-Propyl alcohol, respectively.

### Optical Characterization

The incident light source was a tunable telecommunication laser (santec TSL-550), the wavelength of which could be tuned from 1480 nm to 1630 nm. Light first passed through the polarizer, lens and beam splitter, and then focused on the rear focal plane of the objective lens (10X), to make sure the incident light on the sample was close to normal incidence. The reflected signal was collected by a photodiode (PDA10DT-EC), which could be switched to a CCD to locate the sample. A lock-in amplifier with the help of chopper (SR540) was connected to the photodiode to pick out the signal from the background noise. A pair of orthogonal polarizers were placed at the front and the middle part of the light path, to introduce crossed-polarization which could efficiently decrease the impact from background reflection and promote the relative intensity of the signal.

## Data availability

The data that support the findings of this study are available from the corresponding author upon request.

## Code availability

The code used in this work is available upon request from corresponding author.

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

## Acknowledgements

L.H. and A.E. Miroshnichenko were supported by the Australian Research Council Discovery Project (DP200101353) and the UNSW Scientia Fellowship program. L.H. was also supported by Shanghai Science and Technology Committee (22PJ1402900). C.Z. was supported by the National Natural Science Foundation of China (Grants No. 12004084 and 12164008), Guizhou Provincial Science and Technology Projects (ZK[2021]030). R.J., G.L., and X.C. were supported by National Natural Science Foundation of China (62222514, 62204249, 61991440); Youth Innovation Promotion Association of Chinese Academy of Sciences (Y2021070); Strategic Priority Research Program of Chinese Academy of Sciences (XDB43010200); Shanghai Rising-Star Program (20QA1410400); Shanghai Science and Technology Committee (23ZR1482000, 20JC1416000 and 22JC1402900), Natural Science Foundation of Zhejiang Province (LR22F050004), and Shanghai Municipal Science and Technology Major Project (2019SHZDZX01). A.O. and A.A. were supported by the Air Force Office of Scientific Research and the Simons Foundation. This work was partially carried out at the Center for Micro and Nanoscale Research and Fabrication in University of Science and Technology of China.

## Author contributions

L.H. and A.E.M. conceived the idea. L.H. performed the theoretical calculation and numerical simulation. C.Z. fabricated the sample and helped with numerical simulations. R.J. and G.L. carried out the morphology characterization, built up the optical system and implemented the measurement. A.O., L.X., and F.D. helped with the numerical simulation. X.C., W.L., A.A., and A.E.M. supervised the project. L.H. and A.E.M. prepared the manuscript with input from all authors.

## Competing interests

The authors declare no competing interests.
