## [Peer Review File · Nature Communications]

Ultrahigh-Q Guided Mode Resonances in An All-dielectric MetasurfaceEditorial Note: This manuscript has been previously reviewed at another journal that is not operating a transparent peer review scheme. This document only contains reviewer comments and rebuttal letters for versions considered at Nature Communications.

REVIEWER COMMENTS

Reviewer #1 (Remarks to the Author):

Please find attached the comments.

In this manuscript, Huang et al. present the realization of high Q-factor resonators using a metasurface patterned over SOI to excite guided-mode resonances. The authors report the experimental realization of up to 2.39×10^5 Q-factor using their novel design, where the metasurface is made of a patterned photoresist layer. Furthermore, the authors describe tuning the resonance wavelength by modifying the pattern of the photoresist.

Whilst I understand Reviewer 1's comments on how drawing the presented conclusions from leveraging a low index material in the perturbation layer seems trivial, I believe it is important to recognize that the novelty in this work is not in the realization of the physics of the solution, but rather in the demonstration of it through real world material choices that offer a viable opportunity to the industry as a whole to realize such resonators. The authors have communicated this effectively through a concise description in the abstract and an elaborate discussion in the the introduction and conclusion, within revision they have submitted.

I believe this work is of both interest and importance, considering the activity in the development of high Q-factor resonators within the scientific community, and based on the above described novelty, I side with Reviewer 2 in recommending this work to be published in Nature Communications, with **minor revisions**.

Some comments/clarification required can be found below:

1. Could the authors please clarify what the "fitting" Q-factor in figure 2 refers to? This can be clarified in the text of the manuscript.
2. The authors are recommended to numerically demonstrate the design and realization of high Q-factor resonators in the visible. Since it is highlighted as a challenge yet to be addressed in the manuscript, and later briefly discussed by the authors as a problem that simply requires the appropriate selection of core layers, a thorough numerical demonstration is encouraged.

3. On line 297, page 10, the referred figure should be “Fig 4.g” (not 5.g). Are the various spectra normalized to the maximum value in itself? Why does the Q-factor change with the period, as shown in Fig 4.i.?
4. There are a few grammatical and typographical errors in this manuscript that need to be corrected. Some things to check over again are:
 - i. On line 27 of page 1, “...delicate engineering of the topological features...”
 - ii. On line 320, page 11, the letter “t” following the period, “The scale bar in white...”, should be capitalized.
 - iii. Based on Reviewer 2’s comments, the authors moved from describing “proportional to β^{-2} ” to “inversely proportional to β^2 ”. However, the edit on line 139 has it as “inversely proportional to β^{-2} ” which should be corrected to β^2 .
 - iv. The infinity symbol (∞) is often used to show proportionality (\propto). Please correct this (line 164, page 6, for example). It is also sometimes correctly written, as in line 192.
 - v. Is it common to write it as “xoz plane” (like on line 202, page 7)? I have typically always used and seen “x-z plane”. Please verify and update accordingly.Please keep in mind that this is not an exhaustive list and the authors are encouraged to thoroughly proofread their work before submission.
5. In the figures, where applicable, the authors are encouraged to include gridlines to enhance readability of the plots.
6. Please consider adding the schematic describing the measurement setup leveraged for the optical characterization described in the manuscript (instead of the supplementary material).

Reviewer #2 (Remarks to the Author):

This is a very interesting work that could potentially raise considerable interest in a broad community, due to the general interest in high quality-factor (Q-factor) resonators. Recently, many groups have investigated different approaches to realize high-Q resonators, that could be utilized using freely propagating laser beams. However, many of the approaches are non-trivial to realize at arbitrary operation wavelengths, hindering their usefulness in applications (such as the mentioned sensing and filtering applications).

This work demonstrates, both theoretically and experimentally, a simple and very flexible approach to realize metasurface-based ultrahigh-Q-factor ($Q \sim 10^5$) resonators. The idea is intuitive, but to the best of my understanding still novel, and seems to work well. In essence, a metasurface is designed and fabricated on top of a conventional thin-film waveguiding layer (here standard SOI sample), that enables coupling of normally incident far-field radiation into the guided mode resonances (GMRs) associated with the guiding layer. The main novelty here is that since the metasurface only acts as a small perturbation for the system, the Q-factor of the system can be controlled by changing the amount of perturbation. In essence, by making the perturbation weak enough (e.g. by utilizing very thin metasurfaces) the realized Q-factors can be made very high. As already said, this is great piece of work, and could be of interest for the broad readership of Nature Communications, provided that the work is fine-polished by addressing below minor.

1. Page 1, line 54: '...', the largest Q-factor in experiments has been so far limited by fabrication imperfections (i.e., roughness, disorder and non-uniformity) causing increased radiation loss. The nanofabrication process usually involves dry-etching to define these nanostructures, introducing additional surface roughness, imperfect vertical side walls and long-range non uniformity.' Readers should be given some references that verify/clarify these points to them. Please provide references that analyze and discuss the mentioned issues of surface roughness, properties of side walls and long-range uniformity.

2. Page 4, line 138: 'Q values are inversely proportional to β^{-2} ...'. You mean either proportional to β^{-2} , or inversely proportional to β^2 .

3. Page 7, line 247: 'The robustness of such a design strategy enables us to design ultrahigh-Q GMRs at ultraviolet and visible wavelength ranges by choosing appropriate core layers with high-index, including Si₃N₄, GaN, TiO₂.' This is a very valid/important point, but you might benefit of expressing a pinch of humility here. I therefore recommend re-phrasing to milder '...should/could enable..', because some readers might argue that the small feature sizes required to make UV/VIS high-Q-factor structures start limiting the achievable Q-factors. Your demonstrations are near 1550 nm and 1350 nm, and in practise it might be non-trivial to transfer the methodology to shorter wavelengths.

Reviewer #3 (Remarks to the Author):

In this manuscript entitled 'Ultrahigh-Q Guided Mode Resonances in An All-dielectric Metasurface', the authors proposed an effective geometry and its fabrication process to realize a high quality factor (Q-factor) in an all-dielectric metasurface. Nice simulation and experimental data have been demonstrated. The measured Q-factor of 2.39×10^5 is eye-catching. In general, it is a very interesting and meaningful work, but its novelty and impacts are not enough for publication in Nature Communications.

1) My main concern is its physical novelty. The guided mode excited in the proposed structure is normal. Its high Q-factor is originated from the lower refractive index contrast caused by the photoresist layer. I agree with Reviewers 1 and 3 that the physical principle is intuitive and has been reported before. Of course, the proposed scheme makes some progress and achieves a high Q-factor. It is suggested to highlight the key point by combining simulations and theoretical modal analysis.

There are too many parametric scanning simulations in the main text.

2) Secondly, there is a question mark about its application values. The SOI structure has been generally used in silicone-based electronics and photonics. Any new idea should consider its compatibility unless it can make innovative breakthroughs. Meanwhile, what is the exact material of the 330 nm photoresist layer? Although the fabrication process has been given in the Fabrication section, the final chemical component is still confusing. The ZEP520 and the dimethyl benzene are harmful, will the final chemical component be harmful too? This issue is very important for the applications. And, will the final thickness differ from the original 330 nm ZEP?

3) The details should be checked and improved. For example, the abbreviations SOI, NIR, and hBN should be defined at its first appearance in the main text. The units of μm in Line 342 and 358 should be μm . Supplementary Material should be called directly instead of SM without explanation. There should be space characters between the numbers and the units. The decimal points should be consistent for comparison, i.e., 1.0×10^5 and 2×10^5 in Line 299 should be revised to 1.0×10^5 and 2.0×10^5 or 1×10^5 and 2×10^5 .

In general, I am impressed by the high Q-factor realized by the authors. However, the proposed strategy is a kind of technical modification and cannot meet the novelty requirements of Nature Communications.

REPLY TO REVIEWERS' COMMENTS AND A SUMMARY OF THE CHANGES MADE IN THE MANUSCRIPT

Reviewer #1

In this manuscript, Huang et al. present the realization of high Q-factor resonators using a metasurface patterned over SOI to excite guided-mode resonances. The authors report the experimental realization of up to $2.39E5$ Q-factor using their novel design, where the metasurface is made of a patterned photoresist layer. Furthermore, the authors describe tuning the resonance wavelength by modifying the pattern of the photoresist.

Whilst I understand Reviewer 1's comments on how drawing the presented conclusions from leveraging a low index material in the perturbation layer seems trivial, I believe it is important to recognize that the novelty in this work is not in the realization of the physics of the solution, but rather in the demonstration of it through real world material choices that offer a viable opportunity to the industry as a whole to realize such resonators. The authors have communicated this effectively through a concise description in the abstract and an elaborate discussion in the the introduction and conclusion, within revision they have submitted.

I believe this work is of both interest and importance, considering the activity in the development of high Q-factor resonators within the scientific community, and based on the above described novelty, I side with Reviewer 2 in recommending this work to be published in Nature Communications, with **minor revisions**.

Answer: We thank the Reviewer for their positive evaluation and encouraging comments on our work. In the following, we have revised the manuscript according to these comments.

Some comments/clarification required can be found below:

1. Could the authors please clarify what the “fitting” Q-factor in figure 2 refers to? This can be clarified in the text of the manuscript.

Answer: We thank the Reviewer for this valuable comment. The fitting Q-factor is used to demonstrate that the Q-factor follows the inversely quadratic law versus the perturbation parameters α ($Q \propto \alpha^{-2}$). The perturbation parameter can be the thickness of the perturbation layer, the width of top grating and the center of the slit for the compound grating structure.

We have added one sentence to explain the meaning of fitting Q-factor shown in Fig.3 of the revised manuscript, which is highlighted in red fonts on page 7.

2. The authors are recommended to numerically demonstrate the design and realization of high Q-factor resonators in the visible. Since it is highlighted as a challenge yet to be addressed in the manuscript, and later briefly discussed by the authors as a problem that simply requires the appropriate selection of core layers, a thorough numerical demonstration is encouraged.

Answer: We thank the Reviewer for raising this constructive comment. Following the same strategy, we have designed high-Q resonators in the visible. To eliminate loss, a wide band-gap material, such as GaN, TiO₂, and Si₃N₄, shall be chosen. We use GaN as an example to illustrate the designing procedure. For simplicity, the refractive index of GaN is set as $n=2.4$ because GaN has a refractive index ranging between 2.35 and 2.50. The refractive index of the perturbation layer (i.e., ZEP520, PMMA, SiO₂) on the top is set as $n=1.45$. Given that GaN thin film is usually grown on sapphire (Al₂O₃) substrate, we choose Al₂O₃ as substrate with $n=1.77$. We use GaN as the core layer with a thickness $t_{\text{GaN}}=200$ nm, chosen because it sufficiently thick to support a guided mode resonance (GMR) that leaks only to the 0th diffraction order in the substrate. As shown in Fig.R1a, the resonant wavelength of the guided mode shows a linear dependence on the virtual period. Thus, it is easy for us to design a high-Q resonance in visible by choosing the suitable period. For example, if we target a high-Q resonance at 600 nm, the period shall be around 273 nm. After introducing the perturbation grating layer on the top, a high-Q GMR TE31 can be easily

achieved. Fig.R1b-c shows the Q-factor of the TE31 mode vs the thickness of top layer when the width of top grating is $w=140$ nm. It can be clearly observed that Q-factor can be tuned by the thickness of SiO₂ layer. Similarly, one can control the Q-factor of TE31 mode by the width of top grating if the SiO₂ thickness is fixed as $t=40$ nm, as shown in Fig.R1c. In principle, we can design a high-Q GMR at any wavelength in visible by choosing an appropriate period, as shown in Fig.R1a.

Note that the core layer material is not limited to GaN. To show the robustness of such a design strategy, we design a high-Q GMR at 600 nm by using 200 nm Si₃N₄ on SiO₂. The refractive index of Si₃N₄ is set as $n=2.1$. The relevant results are shown in Fig.R1d-f. In principle, the Q-factor can reach infinity if the thickness is very small or the width approaches either zero or period of the grating. In a real situation, it may be challenging to satisfy this. When the width is close to half period, it can be found from Fig.R1c and f that the Q is between 10^3 and 10^4 . This value can be further improved by introducing partially etching SiO₂ layer. The results are shown in Fig.R2. Clearly, the Q-factor of mode TE31 is enhanced by one order of magnitude compared to the case in Fig.R1. The Q-factor can be further improved if a 3D photonic crystal slab is used. Also, this structure is much easier to be realized in real fabrication.

We also added relevant discussions on page 10 in the revised manuscript, highlighted in red fonts. Also, Fig.R1-2 are included as Fig.S11-12 in the supplementary materials.

Fig.R1 **a** Guided mode wavelength versus the virtual period for 200 nm GaN on a sapphire substrate. The dashed line indicates the wavelength at 600 nm. **b** Q-factor of mode TE31 versus the thickness of top SiO₂ grating width $w=140$ nm and $p=273$ nm. **c** Q-factor of mode TE31 versus the width of top SiO₂ grating with $t=40$ nm and $p=273$ nm. **d** Guided mode wavelength versus the virtual period for 200 nm Si₃N₄ on the glass substrate. The dashed line indicates the wavelength at 600 nm. **e** Q-factor of mode TE31 versus the thickness of top SiO₂ grating width $w=140$ nm and $p=320$ nm. **f** Q-factor of mode TE31 versus the width of top SiO₂ grating with $t=40$ nm and $p=320$ nm.

Fig.R2 **a** Q-factor of mode TE₃₁ versus the thickness of top SiO₂ grating with $t_2=120$ nm, $w=140$ nm and $p=273$ nm. The top grating sits on 200 nm GaN on the sapphire substrate. **b** Q-factor of mode TE₃₁ versus the width of top SiO₂ grating with $t=40$ nm, $t_2=120$ nm and $p=273$ nm. **c** Q-factor of mode TE₃₁ versus the thickness of top SiO₂ grating with $t_2=120$ nm, $w=140$ nm and $p=320$ nm. The top grating sits on 200 nm Si₃N₄ on the glass substrate. **d** Q-factor of mode TE₃₁ versus the width of top SiO₂ grating with $t=40$ nm, $t_2=120$ nm and $p=320$ nm.

3. On line 297, page 10, the referred figure should be “Fig 4.g” (not 5.g). Are the various spectra normalized to the maximum value in itself? Why does the Q-factor change with the period, as shown in Fig 4.i.?

Answer: We thank the Reviewer for pointing out this error. We have corrected it in the revised manuscript. Indeed, we normalized the measured reflection to the maximum value in itself to better present the data. The Q-factors change with the period because, for fixed hole size, a larger period indicates a smaller perturbation(duty circle), thus resulting in a larger Q-factor. The overall trend of the measured Q-factor shown in Fig.4i matched this prediction. The slight deviation may be caused by the scattering loss due to imperfect fabrication.

4. There are a few grammatical and typographical errors in this manuscript that need to be corrected. Some things to check over again are:

i. On line 27 of page 1, “...delicate engineering of the topological features...”

ii. On line 320, page 11, the letter “t” following the period, “The scale bar in white...”, should be capitalized.

iii. Based on Reviewer 2's comments, the authors moved from describing "proportional to " to "inversely proportional to ". However, the edit on line 139 has it as "inversely proportional to " which should be corrected to .

iv. The infinity symbol (∞) is often used to show proportionality (\propto). Please correct this (line 164, page 6, for example). It is also sometimes correctly written, as in line 192.

v. Is it common to write it as "xoz plane" (like on line 202, page 7)? I have typically always used and seen "x-z plane". Please verify and update accordingly. Please keep in mind that this is not an exhaustive list and the authors are encouraged to thoroughly proofread their work before submission.

Answer: We thank the Reviewer for careful reading and pointing out these errors. We have revised the manuscript accordingly. We also carefully went through the manuscript to eliminate other potential errors.

5. In the figures, where applicable, the authors are encouraged to include gridlines to enhance readability of the plots.

Answer: We thank the Reviewer for this nice suggestion. We have included gridlines for some figures to enhance readability of the plots.

6. Please consider adding the schematic describing the measurement setup leveraged for the optical characterization described in the manuscript (instead of the supplementary material).

Answer: We thank the Reviewer for their constructive comments. We have moved the Figure that illustrates the setting up of optical characterization into the main text. This figure is Fig.5 in the revised manuscript.

Reviewer #2 (Comments for the Author):

This is a very interesting work that could potentially raise considerable interest in a broad community, due to the general interest in high quality-factor (Q-factor) resonators. Recently, many groups have investigated different approaches to realize high-Q resonators, that could be utilized using freely propagating laser beams. However, many of the approaches are non-trivial to realize at arbitrary operation wavelengths, hindering their usefulness in applications (such as the mentioned sensing and filtering applications).

This work demonstrates, both theoretically and experimentally, a simple and very flexible approach to realize metasurface-based ultrahigh-Q-factor ($Q \sim 10^5$) resonators. The idea is intuitive, but to the best of my understanding still novel, and seems to work well. In essence, a metasurface is designed and fabricated on top of a conventional thin-film waveguiding layer (here standard SOI sample), that enables coupling of normally incident far-field radiation into the guided mode resonances (GMRs) associated with the guiding layer. The main novelty here is that since the metasurface only acts as a small perturbation for the system, the Q-factor of the system can be controlled by changing the amount of perturbation. In essence, by making the perturbation weak enough (e.g. by utilizing very thin metasurfaces) the realized Q-factors can be made very high. As already said, this is great piece of work, and could be of interest for the broad readership of Nature Communications, provided that the work is fine-polished by addressing below minor.

Answer: We thank the Reviewer for the positive and encouraging comments. In the following, we have revised the manuscript according to these comments.

1. Page 1, line 54: ‘... , the largest Q-factor in experiments has been so far limited by fabrication imperfections (i.e., roughness, disorder and non-uniformity) causing increased radiation loss. The nanofabrication process usually involves dry-etching to define these nanostructures, introducing additional surface roughness, imperfect vertical side walls and long-range non uniformity.’ Readers should be given some references that verify/clarify these points to them. Please provide references that analyze and discuss the mentioned issues of surface roughness, properties of side walls and long-range uniformity.

Answer: We thank the Reviewer for this constructive comment. We have added some relevant references which discussed the effect of fabrication imperfection (i.e., roughness, imperfect vertical side walls, disorder and non-uniformity) on the Q-factors. These references are also listed as follows

[1] Yang, H. et al. *Effects of roughness and resonant-mode engineering in all-dielectric metasurfaces*. *Nanophotonics* 9, 1401–1410 (2020).

[2] Zhou, C. et al. *Bound States in the Continuum in Asymmetric Dielectric Metasurfaces*. *Laser Photon. Rev.* 17, 2200564 (2023).

[3] Kühne, J. et al. *Fabrication robustness in BIC metasurfaces*. *Nanophotonics* 10, 4305–4312 (2021).

Note that the first reference by Yang et al. demonstrated that surface roughness would broaden the resonant linewidth and reduce the Q-factor. The second work shows that imperfect side walls significantly affect the Q-factor. Even a small angle of side wall (close to 90°) can reduce the Q-factor by almost one order of magnitude. Kühne et al. found that disorder or non-uniformity is also a key factor of broadening the resonance’s linewidth.

2. Page 4, line 138: ‘Q values are inversely proportional to β^{-2} ...’ . You mean either proportional to β^{-2} , or inversely proportional to β^2 .

Answer: We thank Reviewer for pointing out this error. We have changed it to “Q values are inversely proportional to β^2 ...”.

3. Page 7, line 247: ‘The robustness of such a design strategy enables us to design ultrahigh-Q GMRs at ultraviolet and visible wavelength ranges by choosing appropriate core layers with high-index, including Si_3N_4 , GaN, TiO_2 .’ This is a very valid/important point, but you might benefit of expressing a pinch of humility here. I therefore recommend re-phrasing to milder ‘...should/could enable..’ , because some readers might argue that the small feature sizes required to make UV/VIS high-Q-factor structures start limiting the achievable Q-factors. Your demonstrations are near 1550 nm and 1350 nm, and in practise it might be non-trivial to transfer the methodology to shorter wavelengths.

Answer: We thank the Reviewer for this excellent suggestion. We have rephrased the sentence based on this suggestion. Also, we have designed high-Q resonators in the visible. To eliminate loss, a wide band-gap material, such as GaN, TiO_2 , and Si_3N_4 ,

shall be chosen. We use GaN as an example to illustrate the designing procedure. For simplicity, the refractive index of GaN is set as $n=2.4$ because GaN has a refractive index ranging between 2.35 and 2.50. The refractive index of the perturbation layer (i.e., ZEP520, PMMA, SiO₂) on the top is set as $n=1.45$. Given that GaN thin film is usually grown on sapphire (Al₂O₃) substrate, we choose Al₂O₃ as substrate with $n=1.77$. We use GaN as the core layer with a thickness $t_{\text{GaN}}=200$ nm, chosen because it sufficiently thick to support a guided mode resonance (GMR) that leaks only to the 0th diffraction order in the substrate. As shown in Fig.R1a, the resonant wavelength of the guided mode shows a linear dependence on the virtual period. Thus, it is easy for us to design a high-Q resonance in visible by choosing the suitable period. For example, if we target a high-Q resonance at 600 nm, the period shall be around 273 nm. After introducing the perturbation grating layer on the top, a high-Q GMR TE₃₁ can be easily achieved. Fig.R1b-c shows the Q-factor of the TE₃₁ mode vs the thickness of top layer when the width of top grating is $w=140$ nm. It can be clearly observed that Q-factor can be tuned by the thickness of SiO₂ layer. Similarly, one can control the Q-factor of TE₃₁ mode by the width of top grating if the SiO₂ thickness is fixed as $t=40$ nm, as shown in Fig.R1c. In principle, we can design a high-Q GMR at any wavelength in visible by choosing an appropriate period, as shown in Fig.R1a.

Note that the core layer material is not limited to GaN. To show the robustness of such a design strategy, we design a high-Q GMR at 600 nm by using 200 nm Si₃N₄ on SiO₂. The refractive index of Si₃N₄ is set as $n=2.1$. The relevant results are shown in Fig.R1d-f. In principle, the Q-factor can reach infinity if the thickness is very small or the width approaches either zero or period of the grating. In a real situation, it may be challenging to satisfy this. When the width is close to half period, it can be found from Fig.R1c and f that the Q is between 10³ and 10⁴. This value can be further improved by introducing partially etching SiO₂ layer. The results are shown in Fig.R2. Clearly, the Q-factor of mode TE₃₁ is enhanced by one order compared to the case in Fig.R1. The Q-factor can be further improved if a 3D photonic crystal slab is used. Also, this structure is much easier to be realized in real fabrication.

We also added relevant discussions on page 10 in the revised manuscript, highlighted in red fonts. Also, Fig.R1-2 are included as Fig.S11-12 in the supplementary materials.

Fig.R1 a Guided mode wavelength versus the virtual period for 200nm GaN on the sapphire substrate. The dashed line indicates the wavelength at 600 nm. b Q-factor of mode TE₃₁ versus the thickness of top SiO₂ grating width $w=140$ nm and $p=273$ nm. c Q-factor of mode TE₃₁ versus the width of top SiO₂ grating with $t=40$ nm and $p=273$ nm. d Guided mode wavelength versus the virtual period for 200 nm Si₃N₄ on the glass substrate. The dashed line indicates the wavelength at 600 nm. e Q-factor of mode TE₃₁ versus the thickness of top SiO₂ grating width $w=140$ nm and $p=320$ nm. f Q-factor of mode TE₃₁ versus the width of top SiO₂ grating with $t=40$ nm and $p=320$ nm.

Fig.R2 **a** Q-factor of mode TE₃₁ versus the thickness of top SiO₂ grating with $t_2=120$ nm, $w=140$ nm and $p=273$ nm. The top grating sits on 200 nm GaN on the sapphire substrate. **b** Q-factor of mode TE₃₁ versus the width of top SiO₂ grating with $t=40$ nm, $t_2=120$ nm and $p=273$ nm. **c** Q-factor of mode TE₃₁ versus the thickness of top SiO₂ grating with $t_2=120$ nm, $w=140$ nm and $p=320$ nm. The top grating sits on 200 nm Si₃N₄ on the glass substrate. **d** Q-factor of mode TE₃₁ versus the width of top SiO₂ grating with $t=40$ nm, $t_2=120$ nm and $p=320$ nm.

Reviewer #3 (Remarks to the Author):

In this manuscript entitled 'Ultra-high-Q Guided Mode Resonances in An All-dielectric Metasurface', the authors proposed an effective geometry and its fabrication process to realize a high quality factor (Q-factor) in an all-dielectric metasurface. Nice simulation and experimental data have been demonstrated. The measured Q-factor of 2.39×10^5 is eye-catching. In general, it is a very interesting and meaningful work, but its novelty and impacts are not enough for publication in Nature Communications.

Answer: We thank the Reviewer for the positive comments on our work. In the following, we have revised the manuscript according to these comments. We hope that the revised version meets the criteria for publication in Nature Communications.

1) My main concern is its physical novelty. The guided mode excited in the proposed structure is normal. Its high Q-factor is originated from the lower refractive index contrast caused by the photoresist layer. I agree with Reviewers 1 and 3 that the physical principle is intuitive and has been reported before. Of course, the proposed scheme makes some progress and achieves a high Q-factor. It is suggested to highlight the key point by combining simulations and theoretical modal analysis. There are too many parametric scanning simulations in the main text.

Answer: We thank Reviewer for this valuable comment. Indeed, guided mode resonances (GMR) in a dielectric grating or photonic crystal slab have been widely used to realize high-Q factors. However, exciting GMRs with an ultrahigh Q-factor based on these approaches in experiment has remained a challenging task. To date, most of experimental works on high-Q GMRs involve the etching of high-index layer, which usually suffers from the inevitable fabrication imperfection that limits the measured Q-factors. Thus, it is of great necessity to develop a robust yet simple and feasible strategy of realizing GMRs with an ultrahigh Q-factor in the experiment.

In this work, we started by investigating the guided modes of a multilayer waveguide system, and then show that the Q-factor of GMR can be pushed to an ultrahigh value by introducing a low-index perturbation layer on the top of high-index of core layer in the waveguide system. The simple relationship between Q-factor of GMRs and the perturbation parameter β , which is $Q \propto \beta^{-2}$, suggests an easy but straightforward way of realizing a high-Q resonance without breaking symmetry or delicately engineering topological quantities. To obtain an ultrahigh Q-factor in the experiments, we just need to introduce a small perturbation for the whole waveguide system. This can be simply done by defining a thin low-index grating layer on the top of a waveguided system (resist layer photonic crystal slab on 220 nm SOI in this work). It is worth noting that both the core layer and top perturbation layer shall be transparent at targeted wavelengths to eliminate loss. This requirement can be easily satisfied by choosing the appropriate materials. The resonant wavelength can be tuned by the periodicity of perturbation grating structure, as demonstrated in Fig.2 in the revised manuscript. Comparing to the other approaches relying on the carefully topological feature engineering or symmetry broken (ultra-small asymmetry parameters is required) for ultrahigh-Q resonances, our approach is simple, straightforward, and robust. It can be used to experimentally realize an ultrahigh-Q GMR in ultraviolet, visible, near infrared, etc.

We would like to emphasize that the refractive index of core layer plays a critical role in obtaining a high-Q factor. The higher the refractive index, the better the field confinement. This can help to suppress the perturbation parameter. For example, in our work, we choose silicon as core layer material because it is not only transparent in the near-infrared but also has a large refractive index. The high refractive index of a core layer can effectively suppress the perturbation effect of low-index top layer. The larger the refractive index contrast, the better the field confinement and the higher the Q-factor. That is why the measured Q-factor in this work is much higher than the others reported so far which rely on patterning the high-index layer with zero-index contrast. Besides, the fabrication of such a high-Q resonator does not involve etching process, thus eliminating some potential factors (rough surface, imperfect vertical side wall, long-range non-uniformity) that can increase scattering loss. All these contributed to an ultrahigh Q-factor in the experiment.

Furthermore, such a designing strategy is very robust and can operate at arbitrary wavelengths. For instance, if we target a high-Q resonance in visible, it is better to choose GaN, TiO₂ or Si₃N₄ as they have a relatively high refractive index and are transparent in visible (See Fig.R1). Note that the refractive index contrast between GaN (TiO₂ or Si₃N₄) and the top perturbation layer material is lower than that of the silicon and resist layer. To obtain an ultrahigh Q-factor, we can choose to introduce a partially etched low-index layer on the top of the core layer, which can still create an ultrasmall perturbation parameter, as proved in Fig.R2. The discussion is added on page 10 and Fig.S11-12 in the revised manuscript and supporting materials

To highlight the key point of this work, we included the relevant discussion on obtaining the structure parameters for an ultrahigh Q-factor by considering the waveguide dispersion relationship, as shown in Fig.2 in the revised manuscript.

Fig.R1 **a** Guided mode wavelength versus the virtual period for 200 nm GaN on the sapphire substrate. Dashed line indicates the wavelength at 600 nm. **b** Q-factor of mode TE₃₁ versus the thickness of top SiO₂ grating width $w=140$ nm and $p=273$ nm. **c** Q-factor of mode TE₃₁ versus the width of top SiO₂ grating with $t=40$ nm and $p=273$ nm. **d** Guided mode wavelength versus the virtual period for 200 nm Si₃N₄ on the glass substrate. Dashed line indicates the wavelength at 600 nm. **e** Q-factor of mode TE₃₁ versus the thickness of top SiO₂ grating width $w=140$ nm and $p=320$ nm. **f** Q-factor of mode TE₃₁ versus the width of top SiO₂ grating with $t=40$ nm and $p=320$ nm.

Fig.R2 **a** Q-factor of mode TE₃₁ versus the thickness of top SiO₂ grating with $t_2=120$ nm, $w=140$ nm and $p=273$ nm. The top grating sits on 200 nm GaN on the sapphire substrate. **b** Q-factor of mode TE₃₁ versus the width of top SiO₂ grating with $t=40$ nm, $t_2=120$ nm and $p=273$ nm. **c** Q-factor of mode TE₃₁ versus the thickness of top SiO₂ grating with $t_2=120$ nm, $w=140$ nm and $p=320$ nm. The top grating sits on 200 nm Si₃N₄ on the glass substrate. **d** Q-factor of mode TE₃₁ versus the width of top SiO₂ grating with $t=40$ nm, $t_2=120$ nm and $p=320$ nm.

2) Secondly, there is a question mark about its application values. The SOI structure has been generally used in silicone-based electronics and photonics. Any new idea should consider its compatibility unless it can make innovative breakthroughs. Meanwhile, what is the exact material of the 330 nm photoresist layer? Although the fabrication process has been given in the Fabrication section, the final chemical component is still confusing. The ZEP520 and the dimethyl benzene are harmful, will the final chemical component be harmful too? This issue is very important for the applications. And, will the final thickness differ from the original 330 nm ZEP?

Answer: We thank the Reviewer for this valuable suggestion. The material of the 330nm photoresist layer is ZEP520. We first spin coating ZEP520 resist layer on top of SOI, followed by baking 3 mins at 180°C. Then, ZEP resist layer pattern is defined by EBL. After that, the patterned ZEP520 resist layer is developed by dimethyl benzene and iso-Propyl alcohol. Following the developing process, we bake the sample for 2-3 mins to remove the chemical residues. Finally, a ZEP520 photonic crystal slab is formed on top of SOI. Thus, the final chemical component is ZEP520, which is a widely used resist. As long as the researcher follows the safe operation procedure, their effects on the human body are almost negligible. Furthermore, it is worth noting that ZEP520 is used as an example of a low-index layer to introduce perturbation. In principle, we can use other low-index materials such as SiO₂ as the perturbation layer, which can eliminate these chemicals during the fabrication process. Thus, safety issue can be resolved. The final thickness of ZEP520 is around 330nm, which depends on the spin-coating process developed in the lab. Slight deviation from 330nm does not influence our results.

3) The details should be checked and improved. For example, the abbreviations SOI, NIR, and hBN should be defined at its first appearance in the main text. The units of μm in Line 342 and 358 should be μm . Supplementary Material should be called directly instead of SM without explanation. There should be space characters between the numbers and the units. The decimal points should be consistent for comparison, i.e., 1.0×10^5 and 2×10^5 in Line 299 should be revised to 1.0×10^5 and 2.0×10^5 or 1×10^5 and 2×10^5 .

Answer: We thank the Reviewer for pointing out these mistakes. We have corrected them according to these comments, and highlighted them in red fonts in the revised manuscript.

In general, I am impressed by the high Q-factor realized by the authors. However, the proposed strategy is a kind of technical modification and cannot meet the novelty requirements of Nature Communications.

Answer: We thank the Reviewer for these constructive comments. Following these comments, we have revised the manuscript accordingly and hope that it meets the criteria for publication in Nature Communications.

REVIEWERS' COMMENTS

Reviewer #1 (Remarks to the Author):

Thank you for the clarification and corrections. I have no further comments or questions and recommend the authors thoroughly review the work for any final corrections before submission. I find this work interesting and beneficial to the community and therefore recommend it to be published in Nature communications.

Reviewer #2 (Remarks to the Author):

The authors have successfully addressed the concerns I had regarding their manuscript.

Reviewer #3 (Remarks to the Author):

I read the revised manuscript and response letter carefully. The main contribution of this manuscript is the development of high Q-factor resonators, instead of the physical novelty. All my concerns have been addressed in a satisfactory way. I recommend acceptance of the revised manuscript.

REPLY TO REVIEWERS' COMMENTS AND A SUMMARY OF THE CHANGES MADE IN THE MANUSCRIPT

Reviewer #1

Thank you for the clarification and corrections. I have no further comments or questions and recommend the authors thoroughly review the work for any final corrections before submission. I find this work interesting and beneficial to the community and therefore recommend it to be published in Nature communications.

Answer: We thank the Reviewer for recommendation. Also, we have carefully gone through the manuscript to eliminate the potential errors. The revised part is highlighted in redfont.

Reviewer #2 (Comments for the Author):

The authors have successfully addressed the concerns I had regarding their manuscript.

Answer: We thank the Reviewer for the positive evaluation on our work.

Reviewer #3 (Remarks to the Author):

I read the revised manuscript and response letter carefully. The main contribution of this manuscript is the development of high Q-factor resonators, instead of the physical novelty. All my concerns have been addressed in a satisfactory way. I recommend acceptance of the revised manuscript.

Answer: We thank the Reviewer for the positive evaluation and recommendation on our work.